# Group-aware Parameter-efficient Updating for Content-Adaptive Neural Video Compression

## ABSTRACT

Content-adaptive compression is crucial for enhancing the adaptability of the pre-trained neural codec for various contents. Although these methods have been very practical in neural image compression (NIC), their application in neural video compression (NVC) is still limited due to two main aspects: 1), video compression relies heavily on temporal redundancy, therefore updating just one or a few frames can lead to significant errors accumulating over time; 2), NVC frameworks are generally more complex, with many large components that are not easy to update quickly during encoding. To address the previously mentioned challenges, we have developed a content-adaptive NVC technique called Group-aware Parameter-Efficient Updating (**GPU**). Initially, to minimize error accumulation, we adopt a **group-aware** approach for updating encoder parameters. This involves adopting a patch-based Group of Pictures (GoP) training strategy to segment a video into patch-based GoPs, which will be updated to facilitate a globally optimized domain-transferable solution. Subsequently, we introduce a **parameter-efficient** delta-tuning strategy, which is achieved by integrating several light-weight adapters into each coding component of the encoding process by both serial and parallel configuration. Such architecture-agnostic modules stimulate the components with large parameters, thereby reducing both the update cost and the encoding time. We incorporate our GPU into the latest NVC framework and conduct comprehensive experiments, whose results showcase outstanding video compression efficiency across four video benchmarks and adaptability of one medical image benchmark.

## CCS CONCEPTS

• **Computing methodologies** → **Reconstruction**; • **Computer vision problems**;

## KEYWORDS

Neural Video Compression, Content Adaptive Neural Compression, Parameter-Efficient Transfer Learning

**ACM Reference Format:**
Anonymous Submission. 2023. Group-aware Parameter-efficient Updating for Content-Adaptive Neural Video Compression. In *Proceedings of Make sure to enter the correct conference title from your rights confirmation emai (ACM MM' 24)*. ACM, New York, NY, USA, 10 pages. https://doi.org/XXXXXXX.XXXXXXX

**Unpublished working draft. Not for distribution.**

## 1 INTRODUCTION

Video compression, a cornerstone of multimedia computing, has been an essential technology for decades. This importance is driven by the need to efficiently store and transmit an ever-growing variety of video content. However, conventional video compression standards (*e.g.*, H.266/VVC [6]) rely on manually designed modules that may no longer suffice for contemporary requirements. In recent times, Neural Video Compression (NVC) techniques [27] that are optimized end-to-end with advanced neural modules have reached performance levels comparable to those of traditional codecs.

Despite achieving significant success, there are still some limitations to NVC in practical coding applications. The primary challenge is generalization, as most existing NVC models struggle to perform adaptively across different sequential contents. This issue arises because most pre-trained NVC models are optimized for a specific expected rate-distortion (RD) cost within a certain domain (*e.g.*, online video resource [47]), leading to a notable domain gap when applied to cases outside the training domain (*e.g.*, medical volumetric images). To mitigate the issue of such domain gaps, several content-adaptive methodologies [7, 12, 22, 24, 35, 38, 42, 43, 45, 46, 49, 51, 54] have been introduced in the field of neural image compression (NIC). Many approaches involve updating the parameters of the encoder neural modules (*i.e.*, online-updating strategy) to generate quantized compressed features with improved rate-distortion values. This enhancement significantly improves compression performance in testing image scenarios.

Conversely, adopting the direct migration of update strategies from NIC to NVC frameworks still seems impractical. A primary obstacle lies in the evolution of NVC architectures, which now incorporate extensive and complex modules [9, 25–27] with millions of parameters on the encoding side. Some early studies [31] have explored online-updating all parameters on the encoding side mirroring strategies used in content-adaptive NICs. Though it initially yielded performance gains in early light-weight NVC frameworks [32], applying such updates to the intricate architectures of contemporary NVCs will result in prohibitively long encoding time and immense computational demands. Recent investigations [41] have also explored updating solely the motion to circumvent the necessity of optimizing all encoding-side parameters. However, given that motion usually represents just a minor portion of the total bit-rates, confining updates to merely motion components substantially limits the scope for performance enhancements.

Another challenge limiting the effectiveness of content-adaptive NVC is error accumulation. Video compression heavily relies on temporal redundancy. *i.e.*, the compression efficiency of the current frame is greatly dependent on the reconstruction quality of the previously compressed frame. However, the online updating strategy, which aims to optimize the rate-distortion value by minimizing both bit-rate and reconstruction quality simultaneously, can negatively impact the quality of subsequent coding frames

through increased error accumulation. This dynamic underlines the difficulty of achieving both high compression efficiency and maintaining quality across frames.

To tackle the aforementioned challenges, we propose a novel content-adaptive NVC method, termed Group-aware Parameter-efficient Updating (GPU). First, leveraging the breakthroughs in Parameter-Efficient Transfer Learning (PETL) [10, 14, 16] across various vision and language tasks, which demonstrates the efficiency of adapting a pre-trained model to new domains by fine-tuning a minimal set of parameters, our approach incorporates a low-rank adaptation style [17] module, namely encoding-side adapters. These adapters are ingeniously crafted to stimulate specific components within the comprehensive encoding modules. By focusing optimization efforts on these adapters, which constitute a significantly smaller portion of the encoding modules' parameters, we can then streamline optimization and reduce costs dramatically. Moreover, we delve into innovative ways to integrate these adapters into the NVC framework, exploring both serial and parallel configurations. The serial adapter is strategically placed immediately after a module, enhancing its processing, while the parallel adapter is woven into the module through a skip-connection layout, offering a blend of original and adapted processing pathways. This strategic placement of adapters allows for the modification of less than **10%** of the parameters in our encoding module, markedly boosting compression efficiency with minimal modification.

Second, to mitigate error accumulation, we introduce a group-aware optimization strategy. This method diverges from previous methods [3, 31, 41], which concentrated on updating a narrow sequence of consecutive frames. Instead, our approach considers the entire Group of Pictures (GoP) during the updating process. Specifically, when compressing an individual frame, we optimize the entire GoP, which it belongs, to minimize the errors in individual reconstructed frames. Furthermore, to circumvent the significant memory demand that group-based updating could impose on the processing of high-resolution videos, we implement a patch-based GoP updating strategy. It spatially segments each GoP into multiple patch-based GoPs, which will be updated sequentially. In general, this approach ensures that the updates made to this framework do not adversely impact the predictive coding process of other frames within the GoP, thus maintaining the integrity and efficiency of the compression across the entire group.

By integrating two above strategies, we incorporate our GPU into the latest NVC framework [27], significantly improving its coding performance across four most representative video compression benchmarks, HEVC [40] Class B, C, D and E datasets. Moreover, to validate the adaptability of our content-adaptive method in various contents, we also test it on compressing medical volumetric image benchmarks ACDC [5], which serves the most important cardiac dataset with multiple MRI slices. Our framework not only outperforms traditional video codecs, such as H.266/VVC [6], but also surpasses previous content-adaptive NVC methods [31, 41] in performance on video benchmarks and adaptability on medical benchmark. Our contributions can be summarized as follows:

- Introduction of encoding-side adapters inspired by Parameter-Efficient Transfer Learning, optimizing a minimal set of parameters by using low-rank adaptation strategy to adapt pre-trained models to new domains efficiently.

- Implementation of a group-aware optimization strategy to mitigate error accumulation by considering an entire Group of Pictures (GoP) during online updating, ensuring consistent and improved predictive coding across frames.

- Integration of our approach to enhance the coding performance of the latest NVC methods, demonstrating superior results over both traditional video codecs and existing content adaptive NVC methods across four key video and one medical volumetric image compression benchmarks.

## 2 RELATED WORKS

### 2.1 Neural Video Compression

Current methods in neural video compression can be intuitively categorized into two main types: *deep residual coding* and *deep contextual coding*. The first-category approach is inspired by established video compression standards [6, 40], which involves predictive coding (*e.g.*, motion compensation) and the encoding of residual data. A notable advancement in this category is DVC [32], which utilizes Convolutional Neural Networks (CNNs) throughout the traditional residual coding process. Following this, numerous works [4, 8, 19, 21, 29, 48] have enhanced this framework by incorporating more robust modules and sophisticated approaches. For instance, FVC [21] implements all coding operations in the feature space to improve compression efficiency.

In the *deep contextual coding* category [9, 13, 25–28, 30, 34, 39], research extends generative-based neural image compression (NIC) techniques by developing spatio-temporal conditional entropy models that leverage both spatial and temporal contexts. In the early studies, some works [13, 30] have introduced spatial-temporal conditional models by maintaining a latent temporal variable produced from all previous compressed frames. Unlike earlier approaches that rely on accumulating all available information, DCVC family [25–27, 39] introduces a series of methods that employ motion compensation from the consecutive compressed frame to derive temporal context directly, which has been further refined by progressive introducing new methods enhance such as multi-scale temporal context mining [39] or hybrid entropy model [26]. The latest iteration of DCVC_DC has shown to surpass the performance of the conventional video coding standard H.266/VVC [6].

Though above methods have achieved notable success, their performances are often constrained across diverse video content. This limitation stems primarily from the fact that many NVC frameworks are pre-trained on specific domains [47], which restricts their generalizability and adaptability to a wide range of content. To address this challenge, this work introduces a content-adaptive NVC method named GPU, which is seamlessly integrated into the latest NVC framework, DCVC_DC. This integration significantly enhances its compression performance, demonstrating the potential of content-adaptive approaches in overcoming the generalizability limitations of existing NVC frameworks.

### 2.2 Content Adaptive Neural Compression

Content-adaptive methods play essential roles in both NIC [7, 12, 22, 24, 35, 38, 42, 43, 45, 46, 49, 51, 54] and NVC [3, 31, 41], which is to refine the latent representation produced by encoder side during encoding process. In this way, the model can adapt to current coding

samples and reduce the domain shift between training and testing domains, which enhances the adaptability when compressing various types of testing data.

Those approaches have yielded significant accomplishments in NIC. For instance, Campos *et al.* [7] refined latent variables through back-propagation, a technique that was further enhanced by Yang *et al.* [49] through the incorporation of a differentiable quantization module. Concurrently, some researchers have pursued decoder-based strategies; for example, Rozendaal *et al.* [43] opted to update all parameters within the decoder and the entropy model. Nonetheless, it's important to note that these decoder-focused methods typically demand higher bit-rates for the storage and transmission of parameters to the decoder.

In the field of NVC, the development of content-adaptive strategies is still in its infancy. Lu *et al.* [33] introduced the first method of this kind by updating all parameters on the encoding side, drawing parallels with most content-adaptive approaches in NIC. This method was seamlessly integrated into early NVC framework [32], which typically featured lightweight architectures. However, as more advanced modules(*e.g.*, Transformers [44]) are incorporated to enhance the performance, modern NVC frameworks [9, 20, 25–28] have become significantly more complex. Updating the entire encoder is now impractical due to the substantial computational load and increased encoding time. To address this issue, Tang *et al.* [41] developed a novel methodology for online and offline updating, specifically targeting the compression of the motion. This approach requires motion information from H.266/VVC [6] as the auxiliary ground-truth. Yet, since motion data represents only a small fraction of the overall bit-rates, optimizing motion alone is insufficient for significant performance gains.

Additionally, a key difference between NIC and NVC lies in video compression's dependence on the reconstruction quality of preceding frames. The compression of any current-coding frame relies heavily on the quality of its reference frame. Updating only a single frame or a small subset, as suggested in certain studies [3, 41], may lead to cumulative errors in subsequent frames.

In contrast, our method employs a group-aware and parameter-efficient updating strategy to overcome the aforementioned challenges. First, we jointly optimize the entire GoP when generating the bitstream for a single frame, which ensures that updating a particular frame does not lead to significant error accumulation. Secondly, we implement an encoder-side adaptor, which allows us to avoid optimizing the entire encoder network with those comprehensive modules. This adaptation makes the updating process more efficient in terms of parameter use.

### 2.3 Parameter-Efficient Transfer Learning

PETL [10] was initially developed to enhance the training efficiency of natural language processing methods. It has since garnered significant attention due to the rapid development of vision and language task [14, 16, 52]. This approach has become increasingly popular as it offers a performance nearly on par with that of full fine-tuning, strategy making it a trendy strategy in contemporary research.

Our PETL strategy is closely aligned with the concept of Low-Rank Adaptation [11, 14, 15, 17, 23, 36, 52], which are recognized for their practicality and efficiency in delta-tuning methods by using adaptors. Various modifications to adaptors have been explored.

These include adjustments in the placement of adaptors within the model [14, 52], the application of targeted pruning techniques to enhance model efficiencys [15], and the adoption of reparametrization strategies to optimize performance[11, 23]. Such innovations demonstrate the adaptability and versatility of adaptors in enhancing task-specific model performance.

To the best of our knowledge, our work represents the pioneering effort to integrate the PETL strategy within the NVC framework. Specifically, we introduce two distinct adaptor configurations: parallel and series adaptors. The parallel adaptor is designed for use with large-parameter modules, such as the motion and contexture feature encoding network. Conversely, the series adaptor is tailored for light-weight modules, including motion estimation, motion encoding, motion hyper-prior encoding and contexture hyper-prior encoding modules. These dual-adaptor styles significantly enhance our ability to optimize the NVC process during encoding.

## 3 METHODOLOGY

### 3.1 GPU

*3.1.1 Patch-based GoP Updating.* To compress a specified video sequence $\mathcal{X} = \{X_1, X_2, \ldots, X_{t-1}, X_t, \ldots\}$, we initially divide it into $K$ non-overlap groups of pictures (GoP), each consisting of $T$ frames, denoted as $\mathcal{G}_k = \{X_t, X_{t+1}, \ldots, X_{t+T-1}\}$. This process results in $\mathcal{X}$ being represented as $\mathcal{X} = \{\mathcal{G}_1, \mathcal{G}_2, \ldots, \mathcal{G}_K\}$. In this work, we adopt the widely used predictive-frame (P-Frame) coding approach in NVC, as referenced in [9, 18, 20, 21, 25–27, 53]. We designate the first frame in each GoP as an intra-coding frame (I-Frame) and the subsequent frames as P-Frames, following a pattern of $\{IPPPP \cdots\}$ and only apply our updating mechanism upon P-frames.

---

**Algorithm 1** Pacth-based GoP Updating

---

**Require:** Data $\mathcal{X} = \{\mathcal{G}_1, \mathcal{G}_2, \ldots, \mathcal{G}_K\}$; Iteration $MAX$; Network $f(\cdot)$; Updating Parameters $\theta_u$; Frozen Parameters $\theta_f$

    **return** The Compressed Data $\hat{\mathcal{X}} = \{\hat{\mathcal{G}_1}, \hat{\mathcal{G}_2}, \ldots, \hat{\mathcal{G}_K}\}$

    **for** $\mathcal{G}_k$ in $\mathcal{X}$ **do**

        **segment** $\mathcal{G}_k$ into $\{\mathcal{G}_k^1, \mathcal{G}_k^2, \cdots \mathcal{G}_k^N\}$

        **for** $STEP$ in $MAX$ **do**

            **for** $\mathcal{G}_k^i$ in $\mathcal{G}_k$ **do**

                $\theta_u^i = argmin_{\theta_u^i} \mathcal{L}(f(\mathcal{G}_k^i, \theta_u, \theta_f), \mathcal{G}_k^i)$

                $\theta_u = \theta_u^i$

            **end for**

        **end for**

        $\hat{\mathcal{G}_k} = f(\mathcal{G}_k, \theta_u, \theta_f)$

    **end for**

---

Our goal is to perform our online updating mechanism, GPU, for each GoP $\mathcal{G}_k$ to minimize error propagation. However, directly updating $\mathcal{G}_k$ poses significant challenges due to the potentially high resolution and large number of frames involved, leading to substantial memory and computational demands. To address this, we spatially partition $\mathcal{G}_k$ into $N$ patch-based GoPs $\mathcal{G}_k^i$, each located at a specific spatial position $i$. This allows us to apply the updating mechanism on each $\mathcal{G}_k^i$ individually on the GPU, using the objective function $\mathcal{L}$, which will be elaborated in *Section* 3.2.2. After updating all patch-based GoPs $\mathcal{G}_k^i$, we proceed to compress the entire GoP $\mathcal{G}_k$ into a bit-stream utilizing our NVC framework with the newly

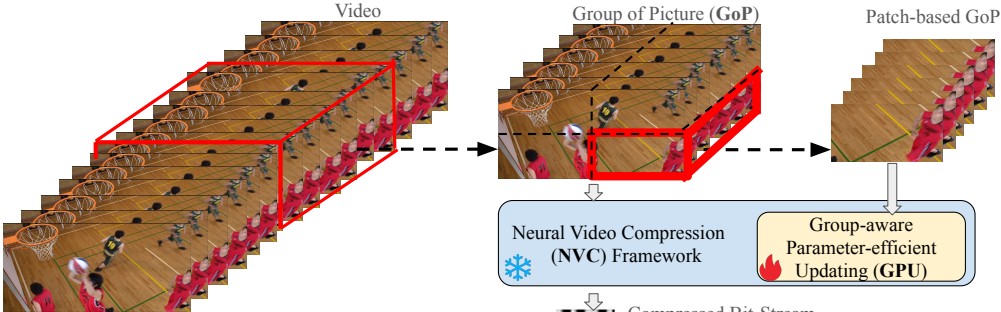

Figure 1: The overview of our Patch-based Groups of Pictures (GoP) updating strategy. It begins by segmenting each video into multiple GoPs as in conventional video codecs. We then spatially divide each GoP into several patch-based GoPs. Following this, we apply our Group-aware Parameter-efficient Updating (GPU) technique to update these patch-based GoPs sequentially. Once all the patch-based GoPs have been optimized, we feed the integrated GoP into the Neural Video Compression (NVC) framework to generate the compressed bit-stream. This process is repeated iteratively until the entire video has been compressed.

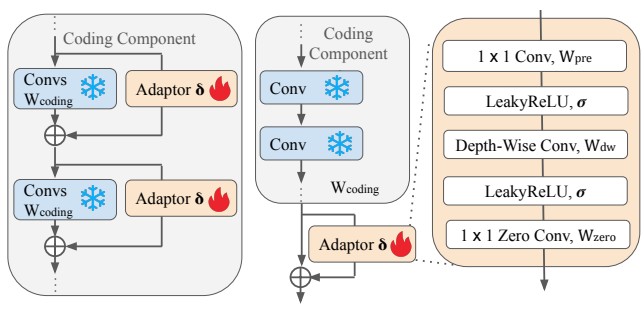

(a) Parallel Configuration    (b) Serial Configuration    (c) Adaptor Architecture

Figure 2: The configurations of how we place our adaptor in (a) parallel and (b) serial ways integrated with our different coding components, and (c) the architecture of our adaptor.

updated parameters on the encoder side. This process is iteratively carried out for all $\mathcal{G}_k$s. We demonstrate this algorithm 1 and its implementation details in Fig. 1.

*3.1.2 Encoder-side Adaptor.* Existing content-adaptive NVC [32, 41] methods often update all parameters in the encoding modules directly. While this full-updating strategy may naturally improve performance, it also significantly increases computational costs. Such a strategy becomes impractical for modern NVC systems [27], which typically involve a vast number of parameters.

To enhance our updating process's efficiency, we have implemented the adaptor mechanism, a technique widely used in PETL methods [17]. As shown in Fig. 2 (c), our approach $\delta(\cdot)$ utilizes a streamlined architecture comprising merely three convolutional layers $\mathbf{W}$ and two LeakyReLU activation layers $\sigma(\cdot)$. This structure is designed to process input features, denoted as $\mathbf{F}_{in} \in \mathcal{R}^{C_{in} \times H_{in} \times W_{in}}$ and generate updated features $\mathbf{F} \in \mathcal{R}^{C \times H \times W}$ using adaptors. Initially, we apply a $1 \times 1$ convolutional layer for pre-processing, characterized by weights $\mathbf{W}_{pre} \in \mathcal{R}^{C_{in} \times C_{dw} \times 1 \times 1}$ to reshape the feature into a channel-squeezed form $\mathbf{F} \in \mathcal{R}^{C_{dw} \times H \times W}$ s.t., $C_{dw} < C_{in}$ & $C_{dw} < C$. Following the strategy commonly employed in Low-Rank Adaptation methods [17, 38], we then apply a depth-wise convolution, with weights $\mathbf{W}_{dw} \in \mathcal{R}^{C_{dw} \times M \times M}$ and a kernel size of $M$ to produce the feature $\mathbf{F} \in \mathcal{R}^{C_{dw} \times H \times W}$. Finally, we employ a $1 \times 1$ (*i.e.*, point-wise) convolutional layer with zero-initialized

weights, $\mathbf{W}_{zero} \in \mathcal{R}^{C_{dw} \times C \times 1 \times 1}$ inspired by [50]. Initially, this results in the delta feature $\delta(\mathbf{F}_{in})$ having no impact on the feature $\mathbf{F}_{in}$. Over the updating process, the impact of $\delta(\mathbf{F}_{in})$ will incrementally increase from zero, optimizing the parameters to enhance the overall model performance. The entire updating can be concisely represented by the following equation, where $\times$ and $\otimes$ denote the convolutional and depth-wise convolutional operations:

$$\delta(\mathbf{F}_{in}) = \mathbf{W}_{zero} \times \sigma(\mathbf{W}_{dw} \otimes \sigma(\mathbf{W}_{pre} \times \mathbf{F}_{in})). \quad (1)$$

The configuration of adaptors plays a crucial role in our approach. Drawing inspiration from previous studies that highlight the importance of strategical adaptors' placement [14, 38, 52], we explore the implementation of adaptors within various coding components in two distinct configurations: parallel and serial placement. Specifically, for complex coding modules (*e.g.*, the contextual feature encoder), that incorporate multiple sophisticated convolutional blocks (*e.g.*, residual blocks) with a large number of parameters, we insert our adaptors in parallel within the coding component. These are placed next to convolutional blocks to generate a delta feature $\delta(\mathbf{F})$. This setup is illustrated in Fig. 2 (a), where the adaptors work alongside convolutional blocks to enhance the feature representation. It can be summarized as:

$$\mathbf{F} \leftarrow \delta(\mathbf{F}) + \mathbf{W}_{coding} \times \mathbf{F}, \quad (2)$$

where $\mathbf{W}_{coding}$ represents the weights with frozen parameters of coding components from the original NVC framework. Conversely, for lighter coding modules (*e.g.*, the hyper-prior encoder), employing parallel insertion for each internal convolution block may not significantly reduce parameters. Therefore, we opt for a serial placement of the adaptor. This approach involves positioning the adaptor after all the smaller convolutional blocks and outside the coding component. Such a configuration is depicted in Fig. 2 (b), where the adaptor acts sequentially, enhancing the module's output by processing the aggregated features from the preceding convolution blocks. It can be summarized as:

$$\mathbf{F} \leftarrow \delta(\mathbf{W}_{coding} \times \mathbf{F}) + \mathbf{W}_{coding} \times \mathbf{F}. \quad (3)$$

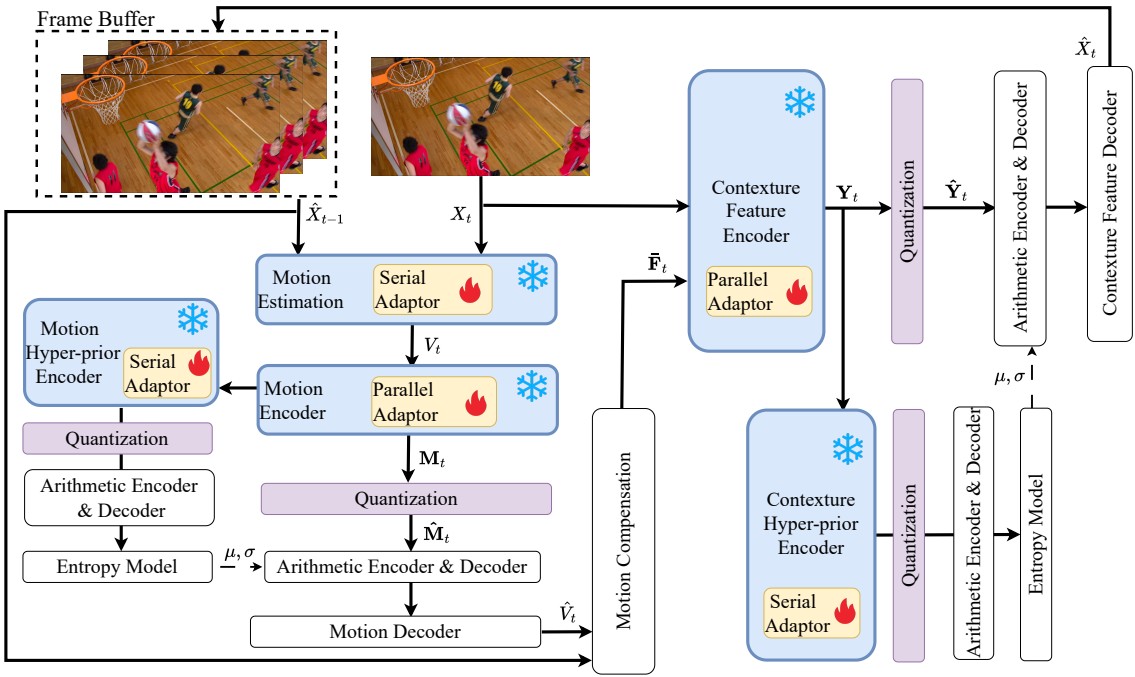

**Figure 3: Overview of our content-adaptive NVC framework with proposed adapters. During online updating, our approach aligns with previous content-adaptive techniques, which only update the modules utilized during the encoding phase (blue). Modules that are also relevant to the decoding phase (white) remain unaltered. The adaptors are integrated within these encoding-specific modules, allowing for the freezing of existing module parameters while updating those within themselves.**

## 3.2 Content Adaptive NVC with Adaptors

*3.2.1 Overview.* We deploy our proposed adaptors in an NVC framework as shown in Fig. 3. Our framework compresses the current frame $X_t$ to obtain the reconstructed frame $\hat{X}_t$ and associated quantized features $\hat{M}_t$ and $\hat{Y}_t$, where the subscript $t$ represents the current time-step $t$. The process involves four main steps: *1. Motion Estimation and Compression*; *2. Motion Reconstruction and Compensation*; *3. Contexture Feature Compression*; and *4. Frame Reconstruction*. In general, we follow the previous content-adaptive NVC methods [33, 41] and only update the modules, that can only be used during the encoding procedure (*i.e.,* in Steps 1 & 3) and leave those modules, which are used in decoding stages (*i.e.,* in Step 2 & Step 4) unaltered. The details are given as follows:

*Step 1. Motion Estimation and Compression.* We first estimate the raw motion $V_t$ between reference frame (*i.e.,* previously reconstructed frame) $\hat{X}_{t-1}$ and current-coding frame $X_t$ through the Motion Estimation operation, where we directly adopt a compact optical flow network SpyNet [37] as in [9, 25–27, 32]. Following this, we employ an auto-encoder-style module, Motion Encoder, to compress $V_t$ into the motion feature $M_t$, followed by quantizing it to $\hat{M}_t$. To losslessly transmit $\hat{M}_t$, we need to adopt arithmetic encoder (*resp.,* decoder) to losslessly encode to (*resp.,* decode from) the bit-stream. To minimize the bit-rates, we employ a hyper-prior-based entropy model as in [27], which enhances the estimation of $\hat{M}_t$'s distribution and improves the efficiency of cross-entropy coding. Consequently, this phase involves generating a quantized motion hyper-prior feature by using a Motion Hyper-Prior Encoder. In this step, we can update the Motion Estimation, Motion Encoder and Motion Hyper-Prior

Encoder online during the encoding procedure, aiming to compress the quantized motion feature $\hat{M}_t$ in a better way. Specifically, we incorporate serial adaptors for those light-weight coding modules, Motion Estimation and Motion Hyper-Prior Encoder, while for more complex coding modules, Motion Encoder, parallel adaptors are employed to facilitate improvements.

*Step 2. Motion Reconstruction and Compensation.* Using the transmitted quantized motion feature $\hat{M}_t$, we can proceed with motion compensation operation. This process begins with the reconstruction of $\hat{M}_t$ into the reconstructed motion $\hat{V}_t$ by employing Motion Reconstruction modules that consist of several standard convolutional blocks. Once we have $\hat{V}_t$, the motion compensation procedure is carried out to generate warped features $\bar{F}_t$, effectively reducing temporal redundancy. In this work, we apply a multi-scale temporal context extraction strategy, directly following [39] to obtain multiple scales of temporal context information $\bar{F}_t$. Since the modules in this phase are integral to both the encoding and decoding stages, we do not update them during the encoding process.

*Step 3. Contexture Feature Compression.* Adhering to the methodologies established by current deep contextual coding frameworks [9, 26, 27, 39], we employ a multi-scale spatial-temporal Contexture Feature Encoder. This encoder generates the latent feature $Y_t$ using the current frame $X_t$ and previously generated warped features $\bar{F}_t$. Similar to Step 1, we again quantize $Y_t$ into $\hat{Y}_t$ and subsequently transmit it losslessly using arithmetic coding alongside a hyper-prior-based entropy model. Consequently, a Contexture Hyper-Prior Feature Encoder is deployed to produce a hyper-prior feature. In this step, we enable the online updating of the Contexture Feature

Encoder and the Contexture Hyper-Prior Feature Encoder. For the more complex Contexture Feature Encoder, parallel adaptors are integrated, whereas for the more compact Contexture Hyper-Prior Feature Encoder, serial adaptors are utilized.

*Step 4. Frame Reconstruction.* Finally, we revert the quantized latent feature $\hat{Y}_t$ back into the reconstructed frame $\hat{X}_t$ by utilizing a multi-scale spatial-temporal Contexture Feature Decoder, with assistance from the warped features $\bar{F}_t$. This step is only used during the decoding process, hence there is no updating of any modules.

*3.2.2 Objective Function.* According to our *Section* 3.1.1, during online updating, we update the parameter $\theta_u^i$ (*i.e.*, parameters of adaptors) by using $argmin_{\theta_u^i} \mathcal{L}(f(\mathcal{G}_k^i, \theta_u, \theta_f), \mathcal{G}_k^i)$. Specifically, we input a patch-based GoP $\mathcal{G}_k^i \in \mathcal{G}_k$ consisting of $T$ patch-based frame $X_t^i \in X_t$ and produce $T$ reconstructed patch-based frame $\hat{X}_t^i$ and associated quantized latent feature $\hat{Y}_t^i$ and quantized motion feature $\hat{M}_t^i$ by using our NVC framework $f(\cdot)$. Consequently, we are now optimizing the objective function $\mathcal{L}$ by solving the following rate-distortion optimization problem:

$$\mathcal{L}(f(\mathcal{G}_k^i, \theta_u, \theta_f), \mathcal{G}_k^i) := \sum_1^T \lambda D(X_t^i, \hat{X}_t^i) + R(\hat{Y}_t^i) + R(\hat{M}_t^i)$$

$$s.t., \hat{X}_t^i, \hat{Y}_t^i, \hat{M}_t^i \leftarrow f([X_t^i, \hat{X}_{t-1}^i, \hat{Y}_{t-1}^i, \hat{M}_{t-1}^i], \theta_u, \theta_f), \tag{4}$$

where $D(\cdot)$ represents the distortion, $R(\cdot)$ represents the bit-rate cost and we use $\lambda$ as a hyper-parameter to control the trade-off between rate and distortion during optimimization.

## 4 EXPERIMENTS

### 4.1 Experimental Protocols

*4.1.1 Datasets.* To evaluate the effectiveness of our approach, we implemented it on video sequences from four of the most representative video compression datasets: HEVC [40] Class B, C, D, and E. These datasets are selected by the standardization committees ISO/IEC and ITU-T and are extensively utilized in the assessment of new video coding technologies. They encompass videos with resolutions ranging from 416×240 to 1920×1080 featuring a diverse array of video scenes. Furthermore, we assessed the adaptability of our method using the most widely used cardiac medical MRI sequence dataset, ACDC [5], where we utilized 10 scans from Patient 101, each with a resolution of $256 \times 232$. We measured the bit-rates by bits-per-pixel (bpp) and assessed the quality of compression by using the peak signal-to-noise ratio (PSNR).

*4.1.2 Baseline Codecs.* Our evaluation aimed to assess the efficacy of our method relative to both traditional video codecs and a range of NVC methods, including standard and content-adaptive variants. For traditional codecs, we utilized the H.265/HEVC [40] and H.266/VVC [6] standards, employing their reference software versions HM-16.20 [1] and VTM-11.2 [2], respectively. Our analysis also included two leading NVCs as baselines: DCVC [25] and its enhanced version, DCVC_DC [27], with the latter representing the latest advancement in NVC study to the best of our knowledge. It surpasses traditional video codecs in performance, sharing the high-level syntax with DCVC but featuring a more intricate architecture. Additionally, we examined two content-adaptive NVC methods: one by Lu *et al.* [33] and another by Tang *et al.* [41] with

Tang's method being the most recent development in the field of content-adaptive NVC to the best of our knowledge. To ensure a fair comparison, we recalibrated the performance metrics for DCVC, DCVC_DC, H.266, and H.265 using publicly available codebases and pre-trained models. Our setup involved an I-Frame interval of 32 and the encoding of 96 frames from each video, consistent with [27]. For each MRI scan from the ACDC dataset, we adopted an I-Frame interval of 30 and encoded 30 slices. For the other codecs, due to the unavailability of public implementations, we relied on performance data as reported in their original studies.

### 4.2 Experimental Results

*4.2.1 Compression Effectiveness on Videos.* In Fig. 4 and Table 1, we present *quantitative comparisons* for rate-distortion and BD-Rate (%) performance of our proposed method compared to existing video compression techniques across four video compression datasets. Our approach demonstrates superior performance over two contemporary content-adaptive NVC frameworks proposed by Tang *et al.* [41] and Lu *et al.* [33], achieving bit-rate reductions of 59.01% and 65.57% on the average. Moreover, our experimental findings reveal a notable average bit-rate savings of 30.69% when compared to the conventional codec H.266/VVC, and a 2.51% improvement over the most recent NVC method, DCVC_DC.

Fig. 6 showcases *qualitative comparisons* among our method, DCVC_DC, and VTM. Notably, our approach achieves superior reconstruction quality, while also utilizing a lower bit-rates for the compression of specific frames (the 80[th] and 57[th]) from the *"BasketballPass"* and *"BlowingBubbles"* sequences. Our method distinctly recovers more structural details, exemplified by the enhanced texture representation in complex areas such as the wall and the tissue box. This level of detail recovery is not observed in the outputs of either DCVC_DC or VTM. Additionally, it has been noted that DCVC_DC tends to introduce blur-related noise artifacts, whereas VTM exhibits ringing artifacts under similar conditions.

*4.2.2 Compression adaptability on Medical Data.* In terms of MRI sequence compression, shown in Fig. 5 on the ACDC benchmark, our method exhibits a marked enhancement in compression performance. Compared to our baseline method, DCVC_DC, and the conventional video coding standard, H.266/VVC, our approach achieves bit-rate savings of 7.85% and 30.05%, respectively. This notable improvement underscores the adaptability of our method, highlighting its capacity to bridge the domain gap, which is a prevalent challenge in current NVC research, thereby allowing us to extend its applicability across varied domains.

*4.2.3 Discussion.* We observed that our method achieves more substantial improvements in medical data than in standard videos. One plausible explanation is that our NVC was pre-trained based on a large-scale video source [47], which closely resembles the content of our test videos, resulting in a smaller domain gap compared to the medical domain. Though performances in the video benchmarks are nearing saturation due to advancements in NVCs, our method demonstrated superior RD values and enhanced perceptual quality. Most notably, the significant improvements in medical data underscore the versatility and broad applicability of our content-adaptive NVC approach, which is crucial for facilitating compression research across diverse data modalities and applications.

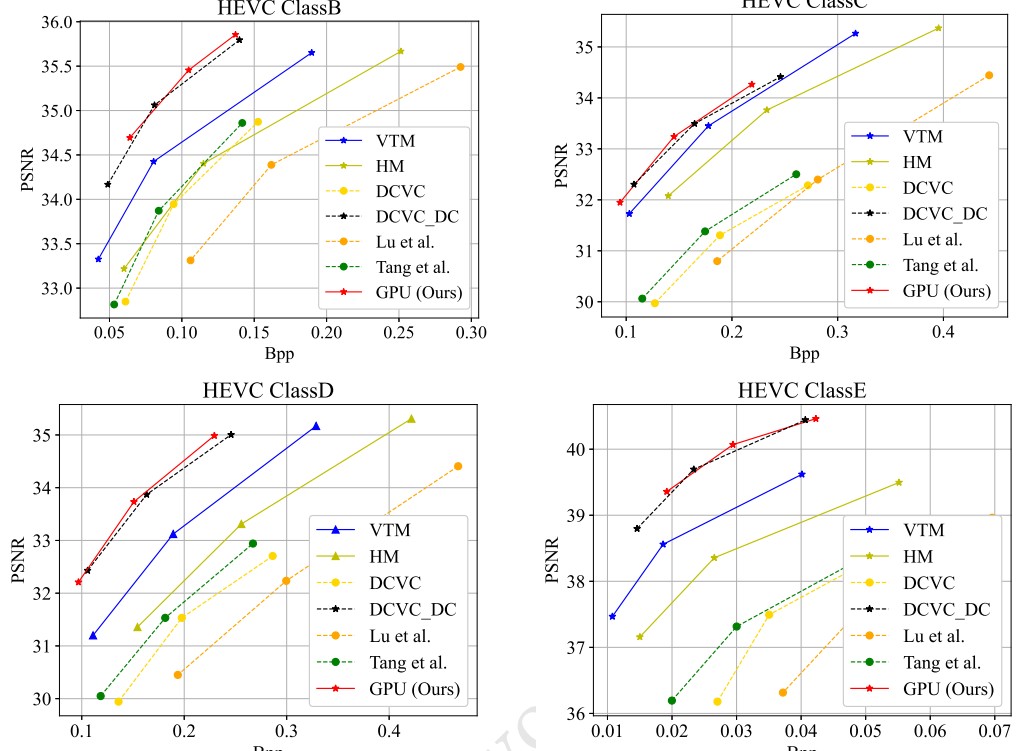

**Figure 4: Rate-distortion (*i.e.*, bit-rates vs PSNR) performance comparison between our and other state-of-the-art video compression methods on the standard video compression benchmarks, HEVC Class B, C, D, and E.**

**Table 1: BD-Rate (%) for PSNR on video benchmark. VTM-11.2 is used as the anchor. Negative values indicate bit-rate savings and positive values indicate additional bit-rate cost. The best and second-best performance are in bold and italic.**

|  | ClassB | ClassC | ClassD | ClassE | Avg |
|---|---|---|---|---|---|
| VTM-11.2 | 0.0 | 0.0 | 0.0 | 0.0 | 0.0 |
| HM-16.20 | 42.86 | 19.51 | 28.42 | 58.76 | 37.39 |
| DCVC | 54.71 | 116.68 | 64.16 | 212.17 | 111.93 |
| DCVC_DC | *-33.00* | *-8.42* | *-29.31* | *-39.03* | *-27.44* |
| Lu *et al.* | 106.54 | 107.61 | 102.56 | 267.96 | 146.17 |
| Tang *et al.* | 43.10 | 94.28 | 48.65 | 199.14 | 96.29 |
| **GPU (Ours)** | **-34.88** | **-12.17** | **-32.13** | **-43.57** | **-30.68** |

## 4.3 Ablation Studies

*4.3.1 Ablation I: Effectiveness of Adaptors.* We undertook an ablation study utilizing the ACDC dataset on the machine with a single NVIDIA A100 GPU and AMD EPYC 7763 CPU. Here, we execute the online updating of all parameters associated with the encoder side (blue components in Fig. 3) within our content-adaptive NVC framework, designated as "Full Updating". Fig. 7 illustrates the RD performance achieved by this full-updating strategy, while Table 2 details the associated computational complexity. Our findings reveal that, although the proposed PETL method incurs a negligible additional bit-rate cost (less than 2%) compared to the full updating approach, they significantly reduce the optimization requirement to less than 10% of the parameters. Moreover, these methods only

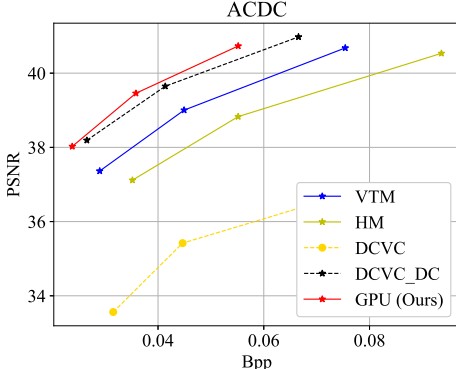

**Figure 5: Rate-distortion (*i.e.*, bit-rates vs PSNR) performance comparison of our and other state-of-the-art video codecs on medical volumetric image compression benchmark, ACDC.**

necessitate 82.1% the optimizing time to train 1 iteration of 1 GoP (*i.e.*, per epoch) required by the full updating strategy. Looking ahead, as more sophisticated techniques are integrated into NVC, it is anticipated that both the number of parameters and the encoding time could further increase, which means our architecture-agnostic updating methods could become increasingly practical.

*4.3.2 Ablation II: Impact of frame number on online-updating efficiency.* We also undertook a comparative analysis on the ACDC dataset to assess the impact of the number of frames updated during the online updating process. Specifically, our default configuration

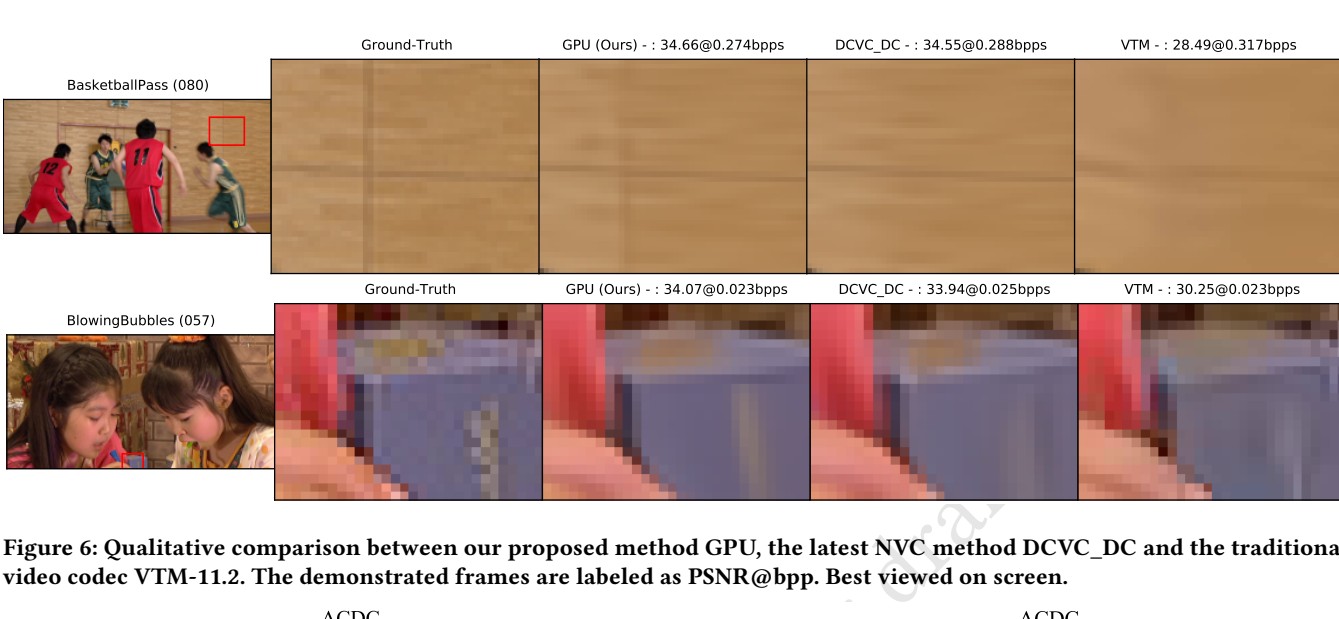

**Figure 6: Qualitative comparison between our proposed method GPU, the latest NVC method DCVC_DC and the traditional video codec VTM-11.2. The demonstrated frames are labeled as PSNR@bpp. Best viewed on screen.**

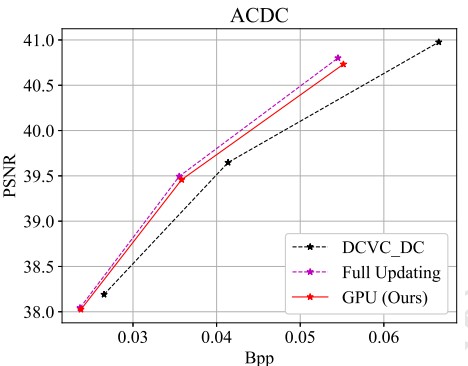

**Figure 7: Performance of our method against the variant updating all encoder-side parameters (*i.e.*, Full Updating).**

**Table 2: The computational complexity on ACDC of our method using encoding-side adapters to delta-tune the encoder (*i.e.*, GPU) against the variant directly optimizing all modules within the encoding network (*i.e.*, Full Updating).**

|                              | Full Updating | GPU (Ours) |
|------------------------------|:-------------:|:----------:|
| #Optimizing Parameters (M)   | 2.91          | 0.23       |
| Optimizing Time (s/Epoch)    | 2.85          | 2.34       |

involves optimizing all frames (*i.e.*, 32) within a GoP in one online updating session. This study extends our framework to include variations in the number of updated frames (*i.e.*, 1, 10, and 20), each subset being smaller than the full GoP size. The results in Fig. 8 reveal that while the alternative strategies, which update only 10 and 20 frames, remain managing to improve upon our baseline method DCVC_DC, our default GPU updating the entire GoP outperforms these alternatives by achieving bit-rate savings of 7.98% and 3.76%, respectively. Notably, the variant only updating single frame experienced a considerable decline in performance, incurring an additional bit-rate cost of 21.86% over our GPU. This significant discrepancy underscores the fundamental differences between content-adaptive NIC and NVC, particularly highlighting the challenges associated with accumulating errors over time.

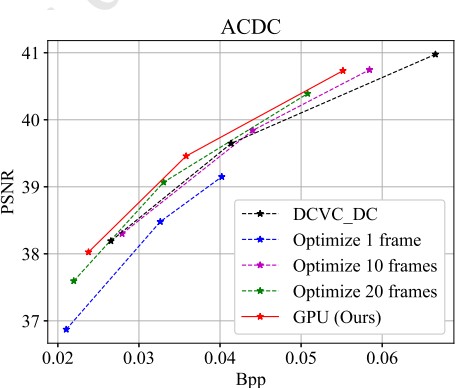

**Figure 8: Performance of our methods against variants optimizing different numbers of frames during online updating.**

## 5 CONCLUSION

In this research, we investigate innovative methods for content-adaptive NVC. We propose a novel strategy, termed GPU, notable for its patch-based GoP updating mechanism and encoder-side adaptor modules, tailored for diverse configurations. Our extensive experiments validate our GPU's effectiveness and adaptability, demonstrating its integration with contemporary NVC framework for the compression of both standard videos and specialized medical MRI sequences. Remarkably, it consistently surpasses state-of-the-art video compression algorithms, such as H.266/VVC and DCVC_DC, thereby establishing a formidable benchmark for both video and MRI compression. These achievements underscore our GPU approach as a robust and efficient baseline for content-adaptive NVC methods. As the landscape of NVC evolves to incorporate more intricate techniques, our architecture-agnostic updating strategy is anticipated to increase, offering a solution to reduce computational requirements while maximizing efficiency. Moreover, such efficient content-adaptive NVC is poised to broaden the scope of NVC technologies, enabling them to address more extensive applications of complex content types beyond standard video. This research heralds new avenues for the compression of more diverse content.

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

Received 20 February 2007; revised 12 March 2009; accepted 5 June 2009

