# OpenReview forum: "Group-aware Parameter-efficient Updating for Content-Adaptive Neural Video Compression"
_acmmm.org/ACMMM/2024/Conference — MM2024 Poster_

### Official Review · Reviewer_KMFd · 2024-05-21

**Rating:** 4
**Confidence:** 4

**Summary:**

This paper presents a novel method for neural video compression (NVC) that adapts to different video contents effectively. The proposed approach, named Group-aware Parameter-efficient Updating (GPU), addresses two main challenges in neural video compression: error accumulation and the complexity of updating large encoder parameters. The GPU method uses a group-aware strategy for parameter updates and introduces lightweight adapter modules for efficient optimization. The method segments videos into patch-based groups of pictures (GoPs) and updates them sequentially, reducing memory demands and improving compression efficiency. The approach is validated on HEVC test sequences, showing superior performance compared to existing methods.

**Strengths:**

1. The use of group-aware parameter-efficient updating and lightweight adapters represents a significant advancement in content-adaptive NVC.
2. Group-aware parameter-efficient updating analyzes the video sequence and adjust GOP size dynamically.
3. The adapter is very light weight, only 3 conv layers and 2 LeakyRelu.
4. A patch-based GoP updating strategy significantly reduces computational complexity and memory requirements.

**Limitations:**

1. The method involves complex segmentation and updating strategies, including online training (adapter) which might be challenging to implement and integrate into existing systems when 2-pass coding is disabled.
2. The process of fine-tuning and adapting the model to new content can be computationally intensive, limiting its use to Random Access mode, it cannot be use in LDB mode as future frames are not available in LDB mode.

**Suitability:**

3

---

### Official Review · Reviewer_a9KL · 2024-05-24

**Rating:** 3
**Confidence:** 4

**Summary:**

The paper introduces Group-aware Parameter-Efficient Updating (GPU), a content-adaptive technique for neural video compression (NVC). It addresses challenges in NVC, such as error accumulation and complex frameworks, by using a patch-based Group of Pictures (GoP) training strategy and a parameter-efficient delta-tuning approach. GPU demonstrates good compression efficiency across various benchmarks.

**Strengths:**

(1) The method is a plug-and-play framework. Based on the theoretical description in the paper, I believe this method can be easily migrated to other neural network video compression frameworks.

(2) The description of GoP updating is very detailed, and I can clearly understand the authors' expression.

(3) Adequate ablation studies demonstrate the effectiveness of Group-aware Parameter-Efficient Updating.

**Limitations:**

(1) I notice that GPU demonstrates a significant performance improvement compared to other content adaptive methods. However, compared to DCVC_DC (baseline), the performance improvement seems less pronounced (30.68% vs. 27.44%). Does it suggest that the architecture of DCVC_DC contributes more to the performance?

(2) When comparing time complexity, I hope the author could include as many methods from Fig. 5 in the comparison as possible.

(3) The datasets tested by authors seem somewhat limited. I hope that testing results on datasets such as MCL-JCV [1] and UVG [2] can be added to demonstrate the generalizability of the method.

[1] Wang, Haiqiang, et al. "MCL-JCV: a JND-based H. 264/AVC video quality assessment dataset." 2016 IEEE international conference on image processing (ICIP). IEEE, 2016.

[2] Mercat, Alexandre, Marko Viitanen, and Jarno Vanne. "UVG dataset: 50/120fps 4K sequences for video codec analysis and development." Proceedings of the 11th ACM Multimedia Systems Conference. 2020.

**Suitability:**

2

---

### Official Review · Reviewer_eMuf · 2024-05-24

**Rating:** 4
**Confidence:** 2

**Summary:**

This paper studies the task of content-adaptive video compression. The authors design a new method called GPU to minimize errors on temporal redundancy. Besides, GPU also uses some light-weight adapters to finetune the model efficiently. Extensive experiments show the advantages of the proposed technique.

**Strengths:**

- Overall, this paper is technical sound.
- The authors also conduct some experiments to show the advantages of their technique over existing baselines.

**Limitations:**

- The number of compared methods is too small. DCVC and DCVC-DC should be included.
- Compression latency, an important metric, should also be reported and compared.
- The continuity of the two motivations in the author's paper does not seem consistent. I think the two novelties are relatively independent. The authors should express the connection between them more accurately.
- The writing and the presentation of the paper should also be advanced. The authors should include an illustration in the 1st or 2nd page to highlight the limitations of existing techniques and their design to address the limitations.

By taking these limitations into consideration, my initial score before rebuttal is a borderline.

**Suitability:**

2

---

### Official Review · Reviewer_YNz2 · 2024-05-25

**Rating:** 3
**Confidence:** 3

**Summary:**

This paper introduces a novel technique called Group-aware Parameter-Efficient Updating (GPU) to enhance neural video compression (NVC). The GPU method employs encoding-side adapters inspired by Parameter-Efficient Transfer Learning (PETL) to stimulate specific components within encoding modules, enabling efficient parameter optimization. These adapters allow the method to efficiently handle varying content types within videos. To mitigate error accumulation and maintain compression quality, the method updates entire Groups of Pictures (GoPs) rather than individual frames, using a patch-based strategy to manage memory demands. The GPU approach improves video compression efficiency and adaptability, outperforming traditional codecs like H.266/VVC and previous content-adaptive NVC methods in experiments on a medical dataset.

**Strengths:**

1. The Paremeter-Efficient Transfer Learning on video compression tasks is quite interesting and effective.

2. The authors present a variety of experimental results, including qualitative outcomes and ablation studies.

3. The proposed method demonstrates better results on out-of-domain datasets compared to DCVC-DC.

4. The paper presentation is well-structured and easy to understand.

**Limitations:**

1. Some recent NIC papers (cited by the authors) used decoder updates. The authors stated that decoder-focused methods require high bit rates. However, the previous work, e.g., [43] showed a promising result. I expect more justification as to why decoder-side adaptation does not work in NVC.

2.  I hate to say that, but, there is no clear improvement compared to the state-of-the-art methods, such as (DCVC-DC) or DCVC-FM [1]. How does your model compare to DCVC-DC in terms of model size (# parameters), encoding, and decoding speed?

3. For a fair comparison, the authors should have provided the experimental results using full updating variants for video compression tasks.  In Figure 4, from my understanding, the RD-curves of Lu et al., and Tang et al. are from the original paper, and the backbone architecture is quite different (model size is also very different). Hence, I do not think you can make the conclusion that the proposed fine-tuning scheme is more effective than full finetuning.

4. Experimental results on MCL-JCV dataset?

5. Why not apply the proposed technique to DCVC variants?

6. The authors tested the impact of updating the number of frames, including a single frame. They stated that they only utilized P-frame compression, which makes it unclear what is meant by updating a single frame (this could be interpreted as 'not updating P-frames' or 'updating only one P-frame'). Additionally, to demonstrate the strength of patch-based GoP updates, comparing them with a small GoP (e.g., 5) with full resolution would be beneficial.

7.  [43] is an NVC paper, but the authors introduced it as neural image compression (NIC).

[Missing references]
 [1] Li, Jiahao, Bin Li, and Yan Lu. "Neural Video Compression with Feature Modulation." arXiv preprint arXiv:2402.17414 (2024).
 [2] Lv, Yue, et al. "Dynamic Low-Rank Instance Adaptation for Universal Neural Image Compression." Proceedings of the 31st ACM International Conference on Multimedia. 2023.

**Suitability:**

2

---

### Meta-Review · Area_Chair_bXWJ · 2024-06-27

**Recommendation:** Accept (Poster)
**Confidence:** 5

**Metareview:**

Based on the reviews, the paper demonstrates a technically sound and innovative approach to Parameter-Efficient Transfer Learning for video compression, with substantial experimental evidence supporting its effectiveness. The strengths, such as detailed ablation studies and better performance on out-of-domain datasets, are significant. However, there were some limitations, including a lack of comprehensive comparisons with state-of-the-art methods and some missing experimental results, and the main important ones have been addressed in the rebuttal. Considering the overall contributions and the clear presentation of the methodology make a compelling case for acceptance, I recommend acceptance (Poster).